# Improving the Efficiency of Test-Time Search in LLMs with Backtracking

## Abstract

Solving reasoning problems is an iterative multi-step computation, where a reasoning agent progresses through a sequence of steps, with each step logically building upon the previous one to reach a desired conclusion. If the desired solution is not attained, the agent must backtrack and try reasoning chains that are quite different from previous attempts. Though prior work such as test-time search against an outcome verifier can improve performance, most search is done in parallel via Best-of-N reranking, and independently for each attempt at a problem, thus wasting a significant amount of computation in sampling multiple full solutions even beyond the point that is needed. Can we reduce the total amount of computation by sharing information and computation across multiple attempts to a given problem? In this paper, we build a novel approach combining process verifiers that predict likelihoods of success *per step* with preemptive backtracking to maximize performance per generated token. To do this, the PRM can be used to identify where a problematic step in a solution trace is by using the sensitivity of the predictions of the learned verifier and allowing the model to do focused resampling of the problematic portion of a solution. This approach can significantly reduce the amount of computation by leveraging partial computation from previous revisions. To further enhance the computational efficiency of inference, we introduce in-context process supervision, where the verifier is conditioned on the history of revisions that are attempted, reducing uncertainty in the verification decisions and improving the verifier's confidence with each round of backtracking. This framework for iterative backtracking, leveraging in-context process supervision, enables an effective tradeoff between inference and model performance.

## 1 Introduction

Solving reasoning problems in large language models (LLMs) involves drawing inferences, making decisions with incomplete or ambiguous information, and solving problems in a structured, logical manner. Reasoning problems are commonplace in many domains such as code development, where reasoning is needed to understand complex logical structures, debug errors, and anticipate edge cases, and mathematics, where it is vital for proving theorems and solving challenging problems.

One promising direction to enhance response quality of responses in reasoning problems is the strategic use of test-time computation, where a model is given an inference compute budget that it could leverage to improve its solution quality. In fact, prior work such as Snell et al. (2024); Charniak & Johnson (2005); Cobbe et al. (2021) explore different mechanisms for solution generation given a fixed generation budget, exploring search algorithms such as BofN, Beam Search, and Look-Ahead search. In scenarios where an LLM generates a complex sequence, small but consequential mistakes made at intermediate steps can lead to a cascade of errors, rendering the entire sequence incorrect. Traditional search methods, which operate in parallel and start from the beginning of the sequence, fail to efficiently handle these errors, often wasting inference-time compute on parts of the solution that are already correct, thus failing to solve intermediate errors. Is there an approach that can more efficiently leverage inference-time compute for error recovery?

In this work, we propose **backtracking-based iterative refinement** for improving LLM solutions. Instead of starting from the beginning with each new sample, our method identifies problematic steps in the solution trajectory and resamples only the parts of the sequence that need correction.

This adaptive approach not only reduces the computational burden but also allows for more targeted revisions, effectively correcting uncorrelated errors and improving the overall solution quality.

Our method leverages process-based verifiers (PRMs), trained to predict points in the reasoning chain where the model is most likely to make mistakes. The PRM estimates the probability of success of a reasoning chain conditioned on a partial solution, akin to a value function in RL, as shown in by Snell et al. (2024). Using the PRM value function, we can then compute an advantage estimate to assess the relative importance of each step and guide the backtracking process by focusing on the steps that contribute least to the solution quality and selectively resampling them. These set of targeted revisions allow for compute to be spent more efficiently, leading to higher-quality solutions with fewer revisions. We also establish a stopping criterion in this framework for early termination of backtracking after a solution has been found.

Ultimately the success of backtracking relies on how accurate the PRM is on intermediate reasoning steps. If we revise a solution after we encounter an unrecoverable state in a reasoning problem, the like-

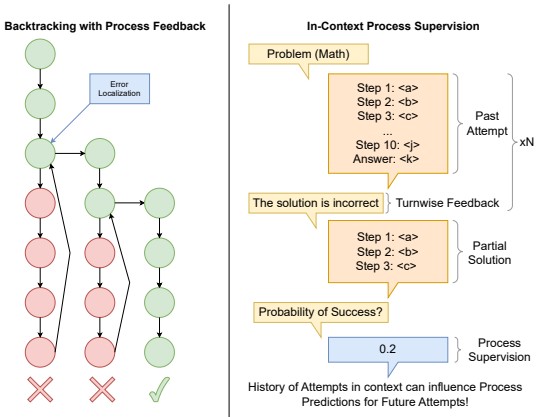

Figure 1: **Left: an illustration of using the value-based verifier for backtracking.** Backtracking is a potential approach to revise an incorrect reasoning chain. Using an error localization criterion, we can identify an appropriate step to revise from to regenerate the solution from. We continue to iteratively revise the solution until a stopping criterion such as satisfying the outcome reward is met. **Right: Leveraging In-Context Process Supervision.** This approach conditions on prior attempts at a problem to reduce the uncertainty of process reward estimates on future attempts.

lihood of success even after recovering is very small. To improve the accuracy of the PRM, we temper distribution shift issues by employing on-policy sampling and label balancing. Additionally, we propose a method to smooth the verifier's outputs to account for estimation errors. We additionally introduce in-context process supervision which allows the PRM to condition on prior attempts at a problem to reduce the uncertainty of the supervision on future attempts. Conditioning on a history of past revisions, the verifier is able to identify failure modes in past attempts and subsequently avoid rating them highly if they are present in future attempts. By providing a framework for sequentially identifying errors and revising them, our approach offers a more scalable and effective solution for test-time inference in LLMs, making it particularly suited for tasks that are more difficult or longer horizon. We demonstrate the computational efficiency of our approach, showing that it significantly improves the test-time compute tradeoff with respect to the number of of generated tokens vs. the accuracy of the generated solution.

## 2 PRELIMINARIES AND NOTATION

A **process reward model (PRM)** assesses the validity of intermediate steps taken during the reasoning process, providing feedback on whether these steps are logical/consistent and make progress towards the solution. Let $\mathbf{s}_t$ represent the state at step $t$, and let $\mathbf{a}_t$ denote the action taken at that step. One approach to learn the PRM is through Monte-Carlo (MC) regression as done in Wang et al. (2023); Snell et al. (2024), where the value at each step is supervised with the Monte-Carlo return-to-go $\mathcal{R}^{\mathcal{D}}(s, a)$, which estimates the probability of success of a rollout from a particular state $s$ and action $a$. The Monte-Carlo return-to-go is computed as the sum of future discounted rewards, $\mathcal{R}^{\mathcal{D}}(s, a) = \sum_{t=i}^{t=T} \gamma^t r_t$, where $\gamma$ is the discount factor, and the sum is taken over the trajectory from the current time step $i$ to the end of the episode. Here, the reward function is sparse, 0 for each intermediate step in the reasoning problem till the answer is predicted. If the answer matches the ground truth answer, a reward of 1 is provided for this final step, otherwise 0. The loss function is divergence of the estimated value from the dataset Monte-Carlo return-to-go:

$$\mathcal{L}_{\mathcal{Q}} = \mathbb{E}_{(s,a)\sim\mathcal{D}} \left[ \mathcal{L}_{KL} \left( Q(s, a), \mathcal{R}^{\mathcal{D}}(s, a) \right) \right] \tag{1}$$

One instantiation of the divergence is soft binary cross-entropy, where each step $(\mathbf{s}_t, \mathbf{a}_t)$ has a target $y_t \in [0, 1]$:

$$\mathcal{L}_{\text{process}} = -\sum_t \left[ y_t \log \hat{y}_t + (1 - y_t) \log(1 - \hat{y}_t) \right], \tag{2}$$

where $\hat{y}_t = \mathcal{R}^{\mathcal{D}}(s, a)$ is the predicted probability that the step $(\mathbf{s}_t, \mathbf{a}_t)$ is correct.

In contrast, an **outcome reward model (ORM)** only evaluates the correctness of the final answer, disregarding individual reasoning steps. Let $\mathbf{o}$ represent the final outcome of the reasoning process. We can similarly learn an ORM through binary classification, where the final outcomes $\mathbf{o}$ are labeled with binary labels $y_{\text{outcome}} \in \{0, 1\}$:

$$\mathcal{L}_{\text{outcome}} = -\sum_i \left[ y_{\text{outcome},i} \log \hat{y}_{\text{outcome},i} + (1 - y_{\text{outcome},i}) \log(1 - \hat{y}_{\text{outcome},i}) \right], \tag{3}$$

where $\hat{y}_{\text{outcome},i} = R_{\text{outcome}}(\mathbf{o}_i)$ is the predicted probability that the final outcome $\mathbf{o}_i$ is correct.

Collecting large amounts of process-level feedback can be undesirable due to noise in the process labels as well as the higher cost of collection. This motivates approximating process-level supervision with a learned value function $Q(s, a)$ can be learned through sparse feedback (outcome rewards).

**Linear search in test-time inference**: Test-time inference often requires efficient search strategies using to navigate the potential solution space. We define **linear search algorithms** as those that operate with a fixed compute budget and a predetermined width of potential completions. Prior work such as Snell et al. (2024) leverages linear search algorithms like Beam Search and Best of N to find near-optimal solutions during inference. In Best of N, $n$ candidate solutions are generated and the solution with the highest evaluation score $S$ is selected:

$$\hat{y} = \arg \max_{y_i \in \{y_1, y_2, \ldots, y_n\}} S(y_i) \tag{4}$$

where $S(y_i)$ denotes the evaluation score of the $i$-th candidate $y_i$, and $\hat{y}$ is the selected output.

In beam search, multiple solution paths are expanded in parallel. At each step $t$, the algorithm retains the top $k$ candidates based on their scores $S(y_{1:t})$:

$$\mathcal{B}_t = \text{Top-}k \left( \{ y_{1:t-1} \cdot y_t \mid y_t \in \mathcal{Y} \}, S(y_{1:t}) \right) \tag{5}$$

where $\mathcal{B}_t$ represents the set of the top $k$ sequences at step $t$, $\mathcal{Y}$ is the set of all possible tokens, and $S(y_{1:t})$ is the evaluation score of the sequence $y_{1:t}$. These methods provide varying trade-offs between computational efficiency and search accuracy, helping to explore the reasoning space effectively during inference.

Linear Search methods can be effective in parallel sampling scenarios, where sequential revisions of a solution are not possible, enabling the discovery of better solutions than single-shot sampling from the model. A **non-linear search algorithm**, in contrast, can adaptively allocate inference time compute. This can allow for more inference time compute to be spent on portions of the problem that are harder to get right. To build a non-linear search algorithm, we will formalize our intuitions in a multi-step single-turn MDP for reasoning, which we describe next.

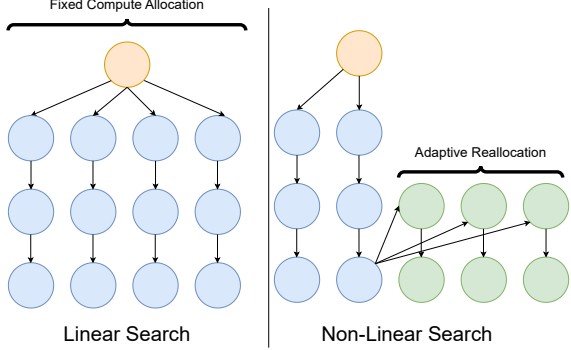

Figure 2: **Linear vs Non-Linear Search** Linear search assumes a fixed compute allocation during inference time. However, with Non-Linear search, we can adaptively allocate inference time compute to parts of the reasoning chain that require more computation.

**Multi-step, Single-turn Markov Decision Process (MDP) for Reasoning**: A reasoning chain in test-time inference can be conceptualized as a multi-step Markov Decision Process (MDP). We formally define a reasoning chain as a trajectory with a bounded horizon $H$, represented as $(s, a, r, t)_0^H$. These trajectories originate from a dataset of prompts $x$ and

responses $y$, which is decomposed into several semantic steps $a_1, a_2, \cdots, a_t$ that, when concatenated, reconstruct the original response $y$.

We formally define an MDP as:

$$\mathcal{M} = (\mathcal{S}, \mathcal{A}, \mathcal{T}, R, \gamma, \rho_0), \tag{6}$$

where:

- $\mathcal{S}$: The state $s_i$ at any step $i$ includes the current prompt $x$ and the sequence of previously selected semantic steps $a_{0\ldots i-1}$.
- $\mathcal{A}$: The action $a_i$ involves selecting the next semantic step in the reasoning chain.
- $\mathcal{T}$: The transition dynamics function is the deterministic concatenation of the state and action: $P(s'|s,a) = \mathrm{concat}(s,a)$.
- $R$: The reward $r = R(s,a)$, which is a function of the current state and action, evaluates the quality or relevance of the selected semantic step within the given context.
- $\gamma$: The discount factor.
- $\rho_0$: The initial state distribution, representing the distribution of initial prompts $x_i$.

This MDP $\mathcal{M}$ allows us to define familiar constructs such as a policy $\pi(a|s)$ and a value function $Q(s,a)$ at a stepwise level.

## 3  BACKTRACKING FOR SEQUENTIAL RESPONSE IMPROVEMENT

As explored in the preliminaries, a linear search algorithm may be a suboptimal formulation for search. To explore this further, consider a didactic problem where an LLM needs to generate a repeating sequence of numbers from 0 to 9. In particular, at each step $t$, the model has 10 possible actions corresponding to the digits 0 through 9, with the correct action determined by the modulus operation $t \mod 10$, which produces the next digit in the sequence. The LLM answer is judged after $N = 1500$ tokens through exact match with the desired sequence. The task is represented in Figure 3.

**Mod 10 Didactic Task**

State (context)    01234567890123

Action (single token)    4

Reward (Outcome)    Gen: 01234567890123..5[1]789
Targ: 01234567890123..5[6]789    } 0

Horizon (N)    1500

Figure 3: **Didactic Task**: Mod 10 sequence generation. The state is the context or the set of characters generated so far. The action is a single character that is generated. The horizon of generation is 1500 characters. The reward is provided at the outcome level, whether the sequence matches the target sequence.

Now, consider a base language model (policy) that is mostly proficient at producing the correct digit at each step but struggles specifically with generating the digit **6** when $t \mod 10 = 6$. Here, the model correctly outputs 6 only 5% of the time and incorrectly outputs 1, leading to significant errors in the sequence. We formally define the base policy as:

$$\pi_{\mathrm{base}}(a_t \mid s_t) = \begin{cases} \Pr(a_t = 6 \mid t \equiv 6 \mod 10) = 0.05 & \text{if } t \equiv 6 \mod 10, \\ \Pr(a_t = 1 \mid t \equiv 6 \mod 10) = 0.95 & \text{if } t \equiv 6 \mod 10, \\ \Pr(a_t = d \mid t \equiv d \mod 10) = 1 & \text{if } t \equiv d \mod 10, d \neq 6. \end{cases} \tag{7}$$

When generating a long sequence of characters $N = 1500$, this error in generating 6 reliably (i.e., whenever $t \mod 10 = 6$) compounds exponentially and results in a substantially wrong sequence. Over 1500 steps, there are 150 opportunities for the model to output the digit 6, but with a 5% success rate, it correctly outputs 6 in only $\approx 8$ instances, leading to 142 errors.

### 3.1  EXAMINING LINEAR AND NONLINEAR SEARCH IN THE DIDACTIC TASK:

Let's examine how a linear search algorithm like Best-of-N would perform in revising this incorrect sequence. If we sequentially apply Best-of-N to revising an incorrect solution, and assume each error is uncorrelated, the expected number of samples required to generate a fully correct sequence becomes **exponential** in the number of errors: $E[N] = 20^{150}$. This requires an impractically

large number of samples to generate a fully correct sequence in expectation, rendering linear search methods ineffective for this task.

Can we instead adaptively utilize our compute resources through **non-linear search**? A common failure mode for linear search is that the search can reach an unrecoverable state (e.g., a mistake in the last steps of a proof), leading to subsequent incorrect steps. By allowing the model to backtrack to a point in the solution before the irrecoverable state, we can give the model an opportunity to try again and correct its errors. Additionally, if multiple uncorrelated errors exist in a solution, this process can be repeated until the problem is solved correctly or the sample budget is exhausted. This adaptive approach prevents wasting computation on parts of the solution that are already correct.

For intuition, let's apply a sequential backtracking procedure to our didactic problem. For purposes of understanding, we start with the simpler scenario where we know the step where a mistake occurs from an oracle. We can backtrack to the step before the mistake and use the same linear search algorithm (Best-of-N) to correct this **single mistake**, which would require $E[N] = 20$ samples in expectation. By performing multiple iterations of revisions and retaining the best solution so far, we can solve the problem in a much more tractable manner, with the total expected number of samples being **linear** in the number of errors: $E[N] = 20 * 150$.

This motivates the need for non-linear search approaches such as backtracking. While parallel algorithms like Best-of-N or Beam Search can perform well in some scenarios, they become computationally expensive when revisions are required, as they necessitate generating a large, fixed number of samples from the beginning of the solution. For challenging or long-horizon problems, this approach is computationally prohibitive, as significant computation could be wasted on generating tokens for parts of the solution that are already correct or occur after an unrecoverable mistake has been made. Next, we will construct a framework for effective backtracking.

## 3.2 Defining the Backtracking Framework for Solution Revision

In this work, we instantiate non-linear search through a novel sequential improvement and backtracking framework for solving reasoning problems. We build a setup to realize a practical implementation of this framework by answering the following key questions: (1) How can problematic parts of a solution be identified and revised?, (2) Can we leverage the history of sequential revisions to better guide search and identify problem parts of a solution, and (3) What are practical considerations to build a robust framework for backtracking? We consider each of these key questions in the following discussion.

**Localizing incorrect steps with PRMs.** The first key component of the framework is identifying where in the solution a mistake is made. One natural choice for this is the **advantage function**, $A^\pi(s, a)$, defined as:

$$A^\pi(s, a) = Q^\pi(s, a) - V^\pi(s) \tag{8}$$

At a particular state (i.e context of a reasoning chain), the advantage function measures the difference in the expected success of a particular action (i.e step in the reasoning chain) compared to a baseline $V^\pi(s)$, or the expected success of actions queried from a proposal distribution or base policy. Intuitively, here the baseline can be viewed as a form of calibration/normalization, where the advantage function not only considers the expected success rate (PRM, $Q^\pi(s, a)$) but the success rate relative to the performance of the proposal distribution or policy.

When the advantage function is low for an action within a trajectory, this indicates that this action may be a poor sample from the proposal distribution in expectation. Therefore, resampling may be desirable as a better action from the proposal distribution can be queried. This motivates using the minimum advantage step within a trajectory $\tau$ to revise from:

$$i_{revise} = \min_i A(s_i, a_i), i \in \{1, \ldots, H\} \tag{9}$$

where $H$ is the horizon of a trajectory $\tau$.

Modeling two different functions, $Q^\pi(s, a)$ and $V^\pi(s)$ is undesirable due to computational inefficiencies in training and querying both functions. One thing we can leverage is that the dynamics of the underlying MDP (as seen in Section 2) is deterministic. Thus, we can choose to model only a Q-value function (PRM), $Q^\pi(s, a)$ and use it to compute $V^\pi(s)$, but querying it at the previous state,

computing the advantage as the value difference between subsequent steps within the trajectory.

$$A^\pi(s', a') = Q^\pi(s', a') - Q^\pi(s, a), \text{ where } s' = concat(a, s), \tag{10}$$

$$i_{revise} = \min_i Q^\pi(s_i, a_i) - Q^\pi(s_{i-1}, a_{i-1}), i \in \{2, \dots, H\} \tag{11}$$

This reformulation allows us to additionally view the advantage as a measure of progress, or how much an action contributes to the success of a trajectory.

**Suffix Generation and Stopping Criteria.** Once the revision step $i_{revise}$ has been identified, we can resample the entire suffix of the solution, conditioned on the partial solution before the minimum advantage step with one of two approaches: (1) Best-Of-N sampling or (2) Beam Search. The value function at the final (solution) step is an outcome verifier. Therefore, we can use the value of the verifier at the final step of the solution to determine which solution to keep. Additionally, we can do this process sequentially, allowing for subsequent revisions from the previously modified reasoning chain(s). We perform this sequential backtracking process a maximum of $M$ times, ensuring that the revision process does not continue indefinitely and we can control the sample budget.

**Summary of the Backtracking Framework** This framework offers an effective approach for solving complex reasoning problems, allowing for robust, incremental improvements that lead to more refined and accurate solutions. We summarize the algorithm framework in Algorithm 1.

### 3.3 ITERATIVE REVISIONS AS A MULTI-STEP, MULTI-TURN MDP.

An additional approach to improving compute efficiency of search is to make the process verifier aware of what did not work well in prior attempts. This should allow the verifier to adapt its predictions over the course of the search process, which intuitively should translate to better value predictions. To do this we need to extend verifiers with some notion of "state" of the search process. This can be done by conditioning the verifier on a linearization of the entire search procedure so far, which we denote as in-context verifiers. This can be formalized as follows:

Sequential Revisions for test-time inference can be viewed as a multi-step and multi-turn Markov Decision Process (MDP). We define a multi-turn revision trajectory $\tau$ as $\tau = \{s, a, r, t\}_0^H$ of a bounded horizon $H$, where each of the $N$ revisions has a sub-horizon (i.e., token lengths) $H_1, H_2, \cdots H_N$. These revision trajectories are constructed from several individual reasoning chains consisting of a prompt $x$ and response $y$. Here, the response is broken into several semantic steps $a_1, a_2, \cdots a_t$, which when concatenated form the original response $y$.

Given a dataset of revision trajectories, we can formally we define a modified MDP $\mathcal{M}_{\text{multiturn}}$ from our Multi-Step, Single-Turn MDP $\mathcal{M}$ (Defined in Section 2) as:

$$\mathcal{M}_{\text{multiturn}} = (\mathcal{S}_{\text{multiturn}}, \mathcal{A}, \mathcal{T}, R, \gamma, \rho_0), \tag{12}$$

where the state $s_i \in \mathcal{S}_{\text{multiturn}}$ at step $i$ in revision turn $k$ consists of the current prompt $x$, the sequence of current turn reasoning steps $a_{0\dots i-1}^{(k)}$, and previous turns reasoning steps $a_{0\dots H}^{(1\dots k-1)}$, turnwise context $c_{0\dots k-1}$.

**Learning In-Context Value Verifiers.** Given the MDP $\mathcal{M}_{\text{multiturn}}$, we can define familiar reinforcement learning objects such as a policy $\pi(a|s)$ and a value function $Q^\pi(s, a)$. These objects can be designed to interact at both a stepwise and turnwise level, enabling a comprehensive supervision process that leverages the history of past revisions and turnwise feedback.

The value function $Q(s, a)$ can be conditioned on both the current reasoning chain and the steps from prior revisions. Let $\{y_1, y_2, \dots, y_{k-1}\}$ represent the set of past $k-1$ revisions, where each revision $y_j$ has its own sequence of steps $\{a_1^{(j)}, a_2^{(j)}, \dots, a_{H_j}^{(j)}\}$. The value function at any step $i$ within the $k$-th revision can then be expressed as:

$$Q(s_{new}, a), s_{new} = (x, a_1^{(1)}, a_2^{(1)}, \dots, a_{H_1}^{(1)}, \dots, a_i^{(k)}, c_{0\dots k-1}, k) \tag{13}$$

Here the turnwise context $c_{0\dots k-1}$ are additional tokens in the form: *Is the turn correct? yes or no.* This token sequence allows the model to understand whether the steps taken in prior revisions were correct or incorrect, providing the value function the outcome of its previous attempts in context.

We can model the cumulative Monte-Carlo return-to-go estimate over all future revisions (given a fixed horizon of revisions), accounting for the potential improvement or deterioration of the solution

as further revisions are made. The cumulative return $\mathcal{R}_t$ for step $t$ within the $k$-th revision is:

$$\mathcal{R}_t = \sum_{i=t}^{H_k} \gamma^{i-t} R(s_i, a_i) + \sum_{j=k+1}^{N} \sum_{o=1}^{H_j} \gamma^{\left[(H_k-t)+\left(\sum_{b=k+1}^{j} H_b\right)+o\right]} R(s_o, a_o) \tag{14}$$

where $H_k$ is the sub-horizon of the current revision, $N$ is the total number of revisions, and $s_o, a_o$ denote the states and actions in future revisions. This cumulative return models the expected future rewards from not just the current revision but also subsequent revisions, allowing the policy to make decisions that optimize long-term performance across all revision stages.

Intuitively, the conditioning of the value function to leverage the historical context provided by prior revisions. This enables models to implement effective strategies such as becoming more confident about a certain mistake, allowing for error correction in subsequent attempts, or more confident about previously successful steps, allowing for further positive reinforcement. In contrast, the single-turn value function would potentially lead to the same deterministic set of failure actions over revision turns as the value function is unable to adapt to previous attempts that the policy has tried.

### 3.4 PRACTICAL CONSIDERATIONS FOR BACKTRACKING

**Tempering Distribution Shift in Process Supervision.** A primary challenge in maintaining the robustness of the value function is the distribution of responses it is trained on. Typically, the value function is initialized using an offline dataset, such as PRM800K, but a significant distribution shift often occurs between the policy's responses during inference and the data used for the initial value function training. This shift can lead to **miscalibration**, where the value function inaccurately assesses the quality of the base policy's step proposals, resulting in sub-optimal outcomes.

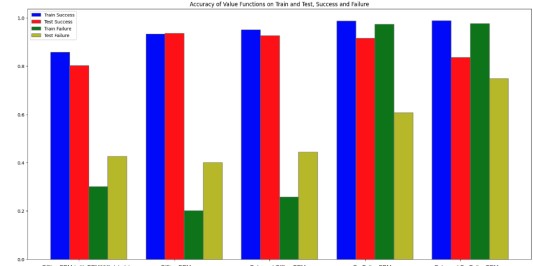

Figure 4: **Outcome Accuracy of PRM**. We evaluate the outcome accuracy of different variants of the process supervision models. Here, both **on-policy** sampling and **label-balancing** leads to higher accuracies for successful and failure trajectories.

The calibration of the value function is critical for effectively guiding the policy model during both backtracking for revision step identification and rollout selection. This calibration becomes particularly important when dealing with steps in backtracking as misidentifying an error can be catastrophic, where an unrecoverable state can be encountered due to errors in the base policy. An unrecoverable state can significantly hinder the success of the solution chain of thought, making it essential to pinpoint where errors occur in the solution process. In particular, the chosen revision step must precede the point where an unrecoverable error is introduced, ensuring that the revision process can correct the trajectory.

To counteract this distribution shift, the PRM is trained with on-policy samples as done in Snell et al. (2024); Luo et al. (2024), allowing the PRM to better identify specific types of errors made by the proposal model during inference.

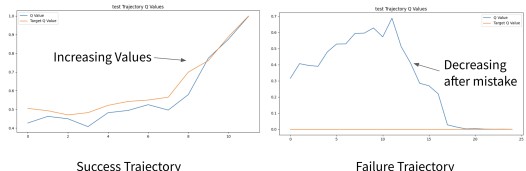

Figure 5: **Learnt PRM Values**: Here we plot the value based PRM for successful and unsuccessful trajectories. For successful trajectories, we see monotonically increasing values that match the target Monte-Carlo Return. For negative trajectories, we see the value increase and then decrease after a mistake is made, allowing us to identify, where to revise from.

Another source of distribution shift is the set of trajectories encountered in revisions being primarily unsuccessful as successful trajectories do not require further revision. Thus, we introduce a label-balancing mechanism during fine-tuning. Specifically, let $\mathcal{D} = \mathcal{D}_{\text{positive}} \cup \mathcal{D}_{\text{negative}}$ be a dataset where $\mathcal{D}_{\text{positive}}$ and $\mathcal{D}_{\text{negative}}$ represent successful and unsuccessful trajectories respectively, and both sets are balanced such that $|\mathcal{D}_{\text{positive}}| = |\mathcal{D}_{\text{negative}}|$. The final objective function for fine-tuning becomes:

$$V_\theta = \arg\min_\theta \left[ \mathbb{E}_{x,a\sim\mathcal{D}_{\text{positive}}} \mathcal{L}(V_\theta(x), \mathcal{R}^{\mathcal{D}_{\text{positive}}(s,a)}) + \mathbb{E}_{x,a\sim\mathcal{D}_{\text{negative}}} \mathcal{L}(V_\theta(x), \mathcal{R}^{\mathcal{D}_{\text{negative}}(s,a)}) \right] \tag{15}$$

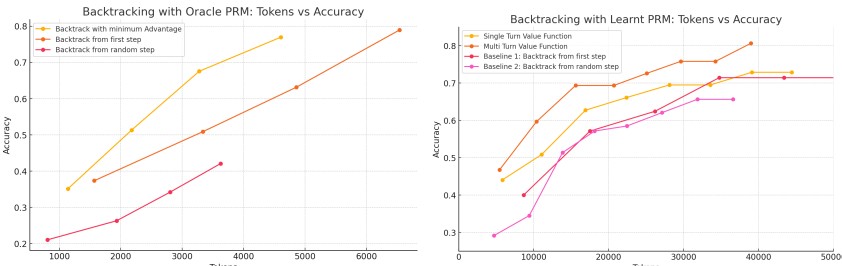

Figure 6: **Backtracking allows for better test time sample efficiency**, measured by generated tokens vs accuracy. Here we allow for up to 4 revisions using the Oracle PRM (**left**) and 8 revisions with the learnt PRM (**right**). We compare against two baselines: revising from the first step and revising from a random step to show the efficacy of localizing errors with the PRM. The choice of suffix generation after a step to revise from is identified is Best-of-N sampling with N=4 for the oracle PRM and N=16 for the learnt PRM.

**Advantage smoothing for effective backtrack step selection.** During inference, estimation errors in computing the advantage function can lead to suboptimal revision step selections. To address this, we introduce a strategy to smooth the value function during the selection process, which we denote as **minimum advantage with tie margin**, where the revision step with the smallest advantage value is selected, but a margin of tolerance is allowed.

Formally, let $A(i) = A(s_i, a_i)$ denote the advantage function for step $i$ in a trajectory $\tau = \{(s_i, a_i)\}_0^H$, and $\omega$ denote the tie margin. We first identify the minimum advantage value $A_{\min} = \min_i A(i)$, and then select the first step $i_{smooth}$ satisfying:

$$i_{smooth} = \min_j \{j : |A_{\min} - A(j)| \leq \omega\} \tag{16}$$

This relaxation allows for a more conservative step to be selected for backtracking, restricting the presence of incorrect steps to be present in the context of the model while decoding.

## 4 EXPERIMENTAL EVALUATION

To evaluate how backtracking would perform on reasoning problems, we consider mathematical reasoning problems from the Hendrycks Math dataset (Hendrycks et al., 2021) for model evaluation. This dataset spans a broad range of mathematical topics, from basic arithmetic to advanced university-level subjects such as algebra, calculus, and geometry. It is designed to assess a model's proficiency in solving both straightforward and complex mathematical problems. We leverage the processed datasets in Sun et al. (2024) which uses outcome/process level supervision from Lightman et al. (2023). To validate the backtracking framework, we explore three key experimental questions.

**Can we more efficiently leverage test-time compute than linear search algorithms?** We first define a metric that captures the balance between inference resources and model performance: the total number of tokens generated across sequential revisions versus the accuracy of the generated solutions in these revisions. The number of generated tokens is a valid element to consider for efficiency as LLM providers such as Together, OpenAI, and Google generally measure cost as a function of both the input and output tokens. Additionally, accuracy is an established metric for performance in reasoning problems such as those found in Hendryks MATH, where a solution can be verified as correct or incorrect by comparing the generated answer with the ground truth answer.

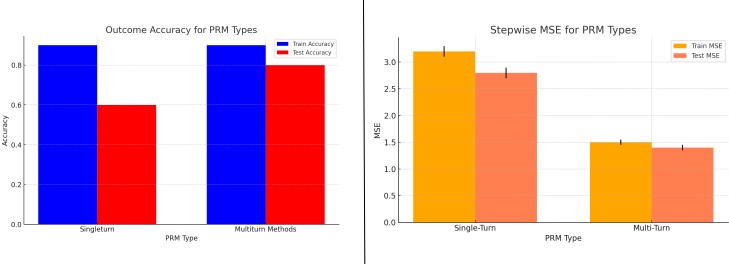

Figure 7: **Outcome and Process Metrics for Value Functions**: We plot the aggregate **left:** outcome accuracy and **right:** process-wise MSE from ground truth Monte-Carlo return-to-go. For both metrics, we find that parameterizing the PRM in the Multi-Turn formulation leads to better performance.

To study sequential revisions, we collect an on-policy dataset utilizing the test-split of Hendryks Math (Hendrycks et al., 2021) with 100 incorrect solutions. From this point sequential revisions

are performed with our framework and two baselines, backtracking to the initial step and random step, to evaluate the success of error localization in our framework. This analysis with two variants of the PRM: an oracle PRM and a learnt PRM as seen in Figure 6. The oracle PRM, estimated by rolling out at each step in the reasoning chain and looking at the success rate by comparing the solution with the ground truth as done in prior work (Wang et al., 2023), provides an upper bound on performance, while the learnt PRM demonstrate the practical effectiveness of our approach. For the oracle PRM, we find that backtracking allows us to see a $\approx 15\%$ improvement over linear search (backtracking from the first step) and significantly outperforms backtracking from a random step, demonstrating the necessity of accurate error localization. We see similar trends with the learned value functions, comparing a single-turn value function, multi-turn (in-context) value function, and baselines of revising from the first and random steps.

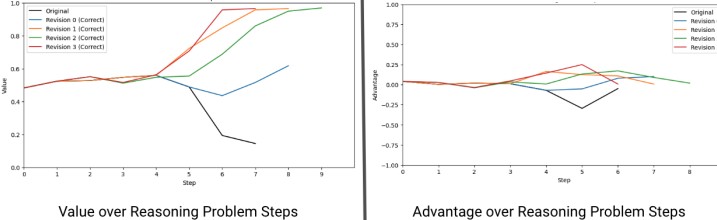

Value over Reasoning Problem Steps      Advantage over Reasoning Problem Steps

Figure 8: **Evolution of Values over Sequential Revisions in a Reasoning Problem**: We plot the evolution of **left:** Value (PRM) and **right:** Advantages for steps in a reasoning problem. In sequential revisions, the value of the trajectory is able to be successfully optimized to be monotonically increasing over steps. The advantage is able to successfully identify a problematic step midway in the trajectory, leading to a successful revisions.

**How effective are the learnt PRMs at localizing errors for backtracking?** One key component of our framework is the learnt process verifier. If the verifier is inaccurate for intermediate steps in the reasoning problem, revisions can be suboptimal, especially if after an unrecoverable state. To evaluate the performance of the PRMs, we consider two metrics: (1) outcome accuracy of the PRM, allowing us to determine if the PRM can correctly identify if a reasoning chain is incorrect/correct, and (2) step-wise MSE from the ground truth monte-carlo return-to-go estimate, to evaluate how the PRM performs on intermediate reasoning steps on held-out queries and steps. We additionally use this metric to compare different parameterizations of our value function such as the turn-independent and turn-dependent PRMs. As seen in Figure 7, the absolute performance of the PRM is high for both variants of the PRM with a high outcome accuracy and low step-wise MSE. Additionally, the in-context, multi-turn PRM can lead to both better outcome accuracy and step-wise MSE. These metrics show the efficacy of the learnt PRM at both an outcome and process level, enabling to use the PRM for both identifying if an error occurred in a reasoning chain and where it did.

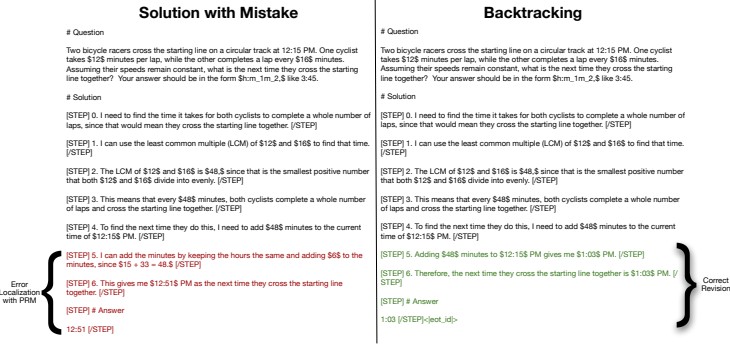

Figure 9: **Qualitative Example of Revision**: In the example revision, the PRM is able to localize where an error is made in the incorrect solution and correct a modular arithmetic mistake.

Futhermore, we qualitatively examine how the behavior of the PRM (value function) evolves over reasoning steps as seen in Figure 5. For correct solutions, the value function should predict monotonically increasing values, while for incorrect solutions, predict lower values after errors occur.

**Do sequential revisions through backtracking exhibit desired behaviors?** We conduct a qualitative analysis of the revision trajectories generated by our backtracking framework. We present a

visualization of the revision trajectories in Figure 9, highlighting in an example reasoning problem in the MATH dataset, an error in modular arithmetic was correctly identified by the PRM.

Additionally, we provide visualizations of the evolution of the learnt value function and corresponding advantage over sequential revisions in Figure 8. The advantage is a useful criterion to identify where a mistake has been made in the trajectory, where resampling just a single revision leads to a successful outcome. Additionally, backtracking enables effective "optimization" over the learnt values, where future revisions have a larger value over reasoning steps than previous revisions.

## 5 RELATED WORK

**Learning and leveraging process-level supervision for LLMs** The idea of using process-level supervision was popularized in Uesato et al. (2022) and more recently in Lightman et al. (2023). Both of these works show the promise of using PRMs for MATH. Building upon this idea, papers such as Math-Shepherd (Wang et al., 2023), MiPS (Wang et al., 2024), and OmegaPRM (Luo et al., 2024) present more efficient automated ways to gather data for process-level rewards.

**Parallel sampling in test-time inference** Test-time inference with search has been extensively studied in works such as Feng et al. (2024); Yao et al. (2023); Hao et al. (2023). One key part of this equation is the way samples are selected (Welleck et al., 2024). A common paradigm involves generating multiple trajectories in parallel, then employing some type of model or function to merge these trajectories. In particular, during test-time inference, the integration of a reward model with a proposal distribution (LLM) can be employed to refine the output responses to a given prompt. For instance, search algorithms such as best-of-N (Charniak & Johnson, 2005) and beam search have been explored in works such as Snell et al. (2024), which leverage reward models to select the most promising candidate samples in reasoning tasks. Another notable family of techniques include self-consistency (Wang et al., 2023) and weighted majority voting (Uesato et al., 2022), which are designed for factual queries with extractable answers. In self-consistency, the language model selects the responses it generates with the highest frequency across multiple samples. This work can be used in conjunction to our approach, without the use of a verifier. More recent methods such as universal self-consistency (Chen et al., 2023) and branch-solve-merge (Saha et al., 2024) explicitly prompt language models to merge the sampled trajectories.

**Iterative revisions in test-time inference** An alternative paradigm to parallel sampling in test-time inference is iterative revisions of reasoning steps. Prior work such as RISE (Qu et al., 2024), SCoRe (Kumar et al., 2024) and Self-Refine (Madaan et al., 2023) parameterize a proposal distribution that can correct mistakes in an incorrect solution. This approach is complementary as our fixed proposal distribution can be substituted with the modified learned distribution from these works. **Additionally, in statistics and Markov chain optimization/inference, Sitewise Resampling and similar approaches (Yang et al., 2019; Wang et al., 2024; Liu et al., 2000; Gagnon et al., 2023) allow for the sequential reuse of generated samples.**

## 6 DISCUSSION, CONCLUSION, AND LIMITATIONS

In this work, we present a framework for sequential response improvement for reasoning problems with PRM based backtracking. Backtracking allows the model to localize where in the response an error has been made and make targeted revisions to efficiently resolve mistakes in a reasoning chain. We additionally introduce in-context process-supervision to allow the verifier to adapt its predictions throughout the search process conditioned on prior attempts at a solution, increasing confidence about a mistake it has made in the past and reinforcing behavior that has led to success. Evaluating this framework with oracle and learned verifiers in the MATH domain, we achieve a $\approx 15\%$ improvement in test-time compute efficiency compared to linear search algorithms.

There are still many open questions and limitations. While we used a fixed proposal distribution for responses, methods like RISE (Qu et al., 2024) leverage self-improvement to steer the distribution. Could these approaches be combined to enhance sequential corrections? Additionally, our analysis focused on a single reasoning domain (math). Can we develop a general verifier applicable across domains such as code, legal reasoning, and robotic planning? How does scaling to multiple domains affect performance, especially in areas where stepwise reasoning is less clearly defined, like code? **Finally, can we use sequential Monte-Carlo sampling approaches to use PRMs effectively?**

## 7 REPRODUCIBILITY STATEMENT

For reproducibility, we provide the following details so readers can replicate the results found in our paper. Firstly, we provide algorithm pseudocode as seen in Algorithm 1, giving the reader transparency in how to replicate the backtracking framework. Additionally, we provide details on how the dataset is curated such as the prompt template as seen in Appendix A.2 and Hyperparameter Details in Appendix A.5. Finally, we provide evaluation details in both the main text in Section 4 and Appendix A.6. For the camera ready, we hope to open-source our on-policy datasets and reward models that we have learned and release a public Github implementation.

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

# A APPENDIX

## A.1 ALGORITHMIC REPRESENTATION FOR BACKTRACKING FRAMEWORK

Below, we provide an algorithmic representation for our backtracking framework for multiple turns of sequential revision, given a fixed test-time budget:

---

**Algorithm 1** Iterative Backtracking and Solution Revision

---

1: **Input:** Initial solution trajectory $\tau = \{s_1, s_2, \ldots, s_N\}$, Maximum iterations $M$, Advantage threshold $\omega$, Success Threshold $p_{des}$
2: **Output:** Revised solution trajectory $\tau'$
3: Initialize iteration count $m \leftarrow 0$
4: Initialize best trajectory $\tau_{\text{best}} \leftarrow \tau$
5: Compute initial advantage function $A(s_i, a_i)$ for each step $i$ in trajectory $\tau$
6: **while** $m < M$ **do**
7:      Identify step $i_{revise} \leftarrow \arg\min_i A(s_i, a_i)$
8:      Apply the smoothing criterion:

$$i_{smooth} = \arg\min_j \left\{ j : |A(s_{i_{revise}}, a_{i_{revise}}) - A(s_j, a_j)| \leq \omega \right\}$$

9:      Resample the suffix of the trajectory starting from step $T = i_{smooth} - 1$
10:      Update the trajectory $\tau' \leftarrow \{s_1, \ldots, s_{T-1}, s'_T, \ldots, s'_N\}$ using a linear search algorithm (e.g Best-Of-N) as a subroutine, conditioning on the prefix of steps prior to index $T$.
11:      **if** Improved Solution Found **then**
12:          Update best trajectory $\tau_{\text{best}} \leftarrow \tau'$
13:      **end if**
14:      Identify last state and action in best trajectory $\tau_{\text{best}}$ as $s_N, a_N$
15:      **if** $Q(s_N, a_N) \geq p_{des}$ **then**
16:          break (stopping criterion)
17:      **end if**
18:      Increment iteration count $m \leftarrow m + 1$
19: **end while**
20: **Return** final revised trajectory $\tau_{\text{best}}$

---

## A.2 DATASET CURATION

We provide additional details in the training datasets used to train the base policy $\pi_{\text{base}}$ and PRM.

### A.2.1 PROMPT TEMPLATE FOR MATH

We use the prompt template in Figure 10 in the training of our PRM and base policy $\pi_{\text{base}}$ (proposal distribution). We add four additional tokens '[STEP]', '[/STEP]', '[TURN]', '[/TURN]' to the vocabulary of our tokenizer that corresponds to the beginning and end of a step or the beginning and end of a revision.

## A.3 OFFLINE DATASETS

For the base policy $\pi_{\text{base}}$ and the offline verifier, we utilize the dataloaders from Sun et al. (2024), which study the PRM800K (Lightman et al., 2023) dataset. Here we use all levels of math problems (1-5) for both our policy and PRM datasets. We construct a Monte Carlo Estimate using the ground truth outcome supervision provided in PRM800K (from Stage 1 + 2).

## A.4 ON-POLICY DATASET COLLECTION

The on-policy dataset was collected using the following approach. For each question, four rollouts were generated with the base policy $\pi_{\text{base}}$ (proposal distribution). These rollouts were then decomposed into partial completions, and 20% of these partial completions were further completed using the current policy and evaluated based on the ground truth reward. In the multiturn setup, each

```
# Question

Find the matrix that corresponds to rotating about the origin by an angle of $45^\circ$
clockwise.

# Solution

[STEP] 0. I know that rotating a point $(x,y)$ about the origin by an angle of $\theta$ clockwise
can be done by multiplying it by a matrix of the form $\begin{bmatrix}\cos\theta & \sin\theta \\
-\sin\theta & \cos\theta \end{bmatrix}$. [/STEP]

[STEP] 1. So I just need to plug in $\theta = 45^\circ$ and simplify. [/STEP]

[STEP] 2. Using the fact that $\cos 45^\circ = \sin 45^\circ = \frac{\sqrt{2}}{2}$, I get that the
matrix is $\begin{bmatrix}\frac{\sqrt{2}}{2} & \frac{\sqrt{2}}{2} \\ -\frac{\sqrt{2}}{2} &
\frac{\sqrt{2}}{2} \end{bmatrix}$. [/STEP]

[STEP] # Answer $\begin{bmatrix}\frac{\sqrt{2}}{2} & \frac{\sqrt{2}}{2} \\ -\frac{\sqrt{2}}{2} &
\frac{\sqrt{2}}{2} \end{bmatrix}$ [/STEP]
```

Figure 10: **Prompt Template for MATH**: The prompt template above is used for the MATH dataset. Each step and revision turn are surrounded by special start and end tokens.

question underwent up to a fixed number of revisions, ranging from 0 to 4. To construct multiturn trajectories, with $K$ revisions and $N$ responses per revision, $N$ perm $K$ potential revision trajectories were considered. Given the large number of possible trajectories, the process was simplified by subsampling $J = 100$ trajectories from the set of $\binom{N}{K}$ combinations to avoid redundancy and manage computational complexity.

A.5    HYPERPARAMETERS FOR VALUE FUNCTION TRAINING + POLICY LEARNING

The base policy $\pi_{\text{base}}$ is initialized with SFT using the following hyperparameters:

| Name | Values |
|---|---|
| Learning Rate (lr) | $1 \times 10^{-6}, 1 \times 10^{-7}$ |
| Schedule | Cosine |
| Warmup Ratio | 10% |
| Model | LLama 3.1 8B Instruct (Dubey et al., 2024) |

Table 1: Hyperparameters used for SFT

The PRM $Q^\pi(s, a)$ is initialized with Monte-Carlo Regression using the following hyperparameters:

A.6    ADDITIONAL EVALUATION DETAILS

We leverage 100 validation queries from the Hendryks Math (Hendrycks et al., 2021) and failure on-policy reasoning chains to construct the dataset for the Performance-Efficiency tradeoff analysis and the evaluation of the learnt PRMs. We ensure that each of the validation queries is unique.

| Name | Values |
|---|---|
| Learning Rate (lr) | $1 \times 10^{-5}, 1 \times 10^{-6}$ |
| Schedule | Cosine |
| Warmup Ratio | 10% |
| PRM Type | Single-Turn, Multi-Turn |
| Model | LLama 3.1 8B Instruct (Dubey et al., 2024) |
| Discount $\gamma$ | 0.8, 0.9 |

Table 2: Hyperparameters used for PRM Training

