# OpenReview forum: "Improving the Efficiency of Test-Time Search in LLMs with Backtracking"
_ICLR.cc/2025/Conference — Submitted to ICLR 2025_

### Official Review · Reviewer_1TMN · 2024-11-03

**Soundness:** 3
**Presentation:** 3
**Contribution:** 1
**Rating:** 3
**Confidence:** 4

**Summary:**

The paper proposes to improve efficiency of test-time search in LLMs with backtracking based on PRMs.

**Strengths:**

The paper correctly identifies a problem with independent monte carlo sampling in high dimensional spaces.

**Weaknesses:**

This is a very old idea, well known in statistics and Markov chain optimization/inference. Sampling the whole trace is called independent sampling. Resampling from a midpoint (whether uniformly or with weighting) is called naive sitewise resampling. Sitewise resampling where there is an attempt to reuse, rather than always resample, the tail is called sitewise resampling with reweighting. This is how sitewise Metropolis Hastings (inference)/ simulated annealing (optimization) works. Reinventing the wheel is often a good idea, including in this case, provided the sources are properly cited. This paper ignores work on sitewise resampling completely.

Performance-wise, I can assume that whoever chose to use 'linear search' for test-time computations in LLMs didn't know  better, but most likely, linear search variants were chosen because they are easily parallelizable. Sitewise resampling is not. However, there are methods that they can both benefit from PRM and are parallelizable. These methods belong to sequential Monte Carlo family. The paper omits time analysis of parallel computations and does not compare wall-clock performance of linear and backtracking search, thus ignoring this problem. If PRM were used to improve the search performance at test time while still preserving advantages of parallel computations, a variant of sequentail Monte Carlo (that is, reweighting and resampling multiple solution based of PRM) should have been worth investigation.

**Questions:**

How are wall-clock times of linear and backtracking search solutions compared, on single queries? Can backtracking search provide interactive experience the way linear search does?

---

> ### Author Response · Authors · 2024-11-21
> **Response to Reviewer 1TMN [1/2]**
>
> We thank the reviewer for their comments and for engaging with the paper. To address the concerns, we revise the related work to discuss sitewise resampling and measure wall-clock time for linear and backtracking search. We are happy to clarify any questions you may have. Please let us know if your concerns are addressed and if so we would be grateful if you would be willing to raise your score. We would be happy to discuss if you have any concerns.
>
> > This paper ignores work on sitewise resampling completely.
>
> Thank you for the references. We have added sitewise resampling to the related works section of the paper. Please let us know if there are any additional sources you would like us to add in this discussion.
>
> > Comparisons based on linear search and sequential Monte-Carlo
>
> Thanks for the pointer to sequential Monte-Carlo methods, we have added and discussed a number of them in the related work section of the paper (in bold). Let us know if there are any sources you would recommend adding or revising in this list.
>
> While we do agree that linear search algorithms with PRMs can be naive and improvements using SMC methods can be made in the long run, **we would like to note, to best of our knowledge, this is the first work that applies ideas similar to sitewise sampling to search with PRMs for improving reasoning.** Snell et al. 2024 present one of the first results that show effective results with beam search on PRMs. Our work pushes the frontier in this space and compares to most existing prior work in this space.
>
> We agree with the reviewer that the community and the authors should look to statistics for literature on non-linear efficient search and apply them to the test-time compute framework. We would like to consider this as future work, highlighting this to the community."

---

> > ### Author Response · Authors · 2024-11-21
> > **Response to Reviewer 1TMN [2/2]**
> >
> > > The paper omits time analysis of parallel computations and does not compare wall-clock performance of linear and backtracking search, thus ignoring this problem.
> >
> > Thanks for the suggestion! It is a great idea to compare wall-clock time for linear and backtracking search, and we now add a new result comparing wall-clock time for our approach in the table below. We utilize the SGLang Engine (Yin et al 2024) as an inference server. We measure this in the code-contests domain and report the results in the table below with the llama 3.1 8b model with a max generation length of 768 tokens and 8 Nvidia A100 gpus (device parallel, round-robin). Here for each comparison, we generate 4 parallel samples and perform 4 sequential revisions for the backtracking algorithm and up to 16 samples for the Best of N. We report the average wall clock time per problem.
> >
> > | N  | Tokens Generated | Wall Clock Time (Parallel Best-of-N) | Wall Clock Time (Sequential Backtracking) |
> > |----|-------------------|--------------------------------------|-------------------------------------------|
> > | 4  | 3072             | 9.07                                |  9.07                                      |
> > | 8  | 6144             | 9.22                                | 17.78                                     |
> > | 12 | 9216             | 9.20                                | 24.84                                     |
> > | 16 | 12288            | 9.33                                | 31.89                                     |
> >
> > As shared, there is an increase in latency for generating more samples sequentially, which is perhaps expected. That said, we do note the focus of this work was to show that backtracking with a PRM can lead to a better tradeoff between accuracy and inference-time token budget, which all of our results attempt to measure. This is akin to motivation from prior work in the community that studies sequential re-sampling of revisions (e.g., RISE (Qu et al. NeurIPS 2024), Snell et al. 2024) and shows that sequential re-sampling is possible but does not investigate accuracy vs token tradeoffs that we also study. To the best of our knowledge, we show some of the first results demonstrating effective accuracy vs token tradeoffs when backtracking against a PRM verifier, improving over past results from Qu et al. NeurIPS 2024 and Snell et al. 2024.
> >
> >  An important point to mention here is that with the incorporation of ideas such as PagedAttention (Kwon et al, 2023) and continuous batching (Daniel et al, 2023), the computational cost of inference does not depend upon the wall-time but the number of tokens generated. We show our approach is more efficient than linear search approaches in number of tokens vs accuracy tradeoff. Of course, parallelizability will be critical factors in large-scale deployment, and even existing techniques based on speculative decoding (Leviathan et al, 2022), prefix caching/ChunkAttention (Ye et al, 2024), and tensor/pipeline parallelism approaches such as Mnemosyne (Agrawal et al, 2024) could be directly used with our approach. That said, this is orthogonal to our goal of demonstrating that backtracking against a PRM can lead to good tradeoff between accuracy and token budgets. Additionally, SOTA reasoning systems such as o1 and DeepSeek R1-Lite leverage large amounts of sequential inference to attain higher accuracy on difficult problems, with models such as DeepSeek R1-Lite generating 10,000 sequential tokens for AIME math problems, resulting in inference taking 200 seconds or longer for each prompt (DeepSeek API).

---

> > > ### Author Response · Authors · 2024-11-24
> > > **Following up!**
> > >
> > > Thank you for your review! Please let us know if further detail is needed or if the new experiments address your concerns.

---

> > > > ### Author Response · Authors · 2024-11-27
> > > > **Following up again!**
> > > >
> > > > Thank you again for your valuable feedback! With the discussion period ending soon, we kindly ask if the additional experiments and clarifications provided address your concerns or if there are any remaining points we can clarify before the deadline.

---

> > > > > ### Comment · Reviewer_1TMN · 2024-12-01
> > > > >
> > > > > I've updated my confidence score based on the authors' feedback.

---

> > > > > > ### Author Response · Authors · 2024-12-01
> > > > > >
> > > > > > We sincerely appreciate the reviewer’s ongoing engagement and thoughtful critique of our work. However, we are somewhat perplexed by the increase in confidence for a lower score, particularly in light of the detailed responses and clarifications provided. Could the reviewer kindly elaborate on any specific concerns that remain unaddressed, so we may further address them?
> > > > > >
> > > > > > Additionally, we feel it is necessary to respond to the perceived weaknesses highlighted by the reviewer, as many of these concerns appear to stem from broad critiques applicable to prior work rather than being specific to our study. We address each point below to clarify and substantiate our position.
> > > > > >
> > > > > > > Sitewise Resampling and Sequential Monte-Carlo approaches
> > > > > >
> > > > > > We acknowledge that sitewise resampling and Sequential Monte Carlo methods allow for the reuse of samples. However, it is important to note that these approaches have not been explored in the context of reasoning with LLMs. In good faith, we have appropriately cited sitewise resampling in the related works section and included a discussion of these methods in our manuscript. Nevertheless, we maintain that these works bear only tangential relevance to the present study. This is evident from their lack of demonstrated applicability to reasoning tasks and the assumptions made about sampling distributions (e.g., Gaussian priors), which diverge significantly from the LLM priors used in reasoning.
> > > > > >
> > > > > > Furthermore, to the best of our knowledge, no prior work has proposed or studied an external verifier guiding resampling with smoothened advantage and in-context value verification. Our methodology uniquely enables the backtracking system to robustly identify errors within reasoning chains and reduce uncertainty during sequential revisions. This represents a novel contribution that, to our understanding, has not been explored within the sitewise resampling paradigm.
> > > > > >
> > > > > > > Latency
> > > > > >
> > > > > > Though latency is an important consideration, there are several approaches, from prior work that we have highlighted above, that can allow for a reduction in latency in production systems. Thus, the latency we measure for the backtracking system is a **significant overestimate** of what latency would look like for a production system, and can be viewed as a conservative upper bound.
> > > > > >
> > > > > > Additionally, the reduction in tokens generated by our approach directly correlates with a reduction in FLOPs, as demonstrated in Sardana et al. (2023). This aligns with established metrics for computational cost in prior studies. Finally, both prior academic work such as RISE (Yu et al, 2024) and Snell et al 2024 as well as SOTA reasoning systems such as O1-preview, Deepseek r1-lite, and Qwen QWQ, similarly incur latency trade-offs due to sequential sampling aimed at improving the performance of the overall reasoning system. Therefore, this is a metric we shouldn’t be penalized for.
> > > > > >
> > > > > > In light of these clarifications and the contextualization of our contributions within existing literature, we respectfully urge the reviewer to reconsider their evaluation of this work. We believe that our methodology and results represent a meaningful advancement in LLM reasoning and address several limitations unaddressed in prior studies.

---

### Official Review · Reviewer_QZQj · 2024-11-04

**Soundness:** 3
**Presentation:** 2
**Contribution:** 2
**Rating:** 6
**Confidence:** 2

**Summary:**

This paper introduces a framework to enhance reasoning tasks using PRM-based backtracking by enabling shared reasoning information and computations across multiple process verifiers. The approach predicts the likelihood of success at each step and identifies problematic steps. By incorporating in-context process supervision, the framework reduces uncertainty in verification decisions, significantly improving inference efficiency.

**Strengths:**

1. I like the idea that identifying problematic steps in the reasoning chain and resample only those parts, rather than starting over with each new attempt. This adaptive approach reduces computational load, allowing for more targeted corrections that effectively address irrelevant errors and enhance the overall quality of the framework.
2. Backtracking in the framework allows the model to locate the root cause of reasoning error. In-context process-supervision allows the model to adapt the prediction results during the search process and increase the confidence about the learning results.
3. Evaluation in the MATH domain shows that this approach improves test-time computational efficiency by 15% compared to the linear search algorithm.

**Weaknesses:**

1. The dataset is limited to MATH domain.
2. The words in the figures are too small to read. The writing needs to be improved.

**Questions:**

Why was a linear search algorithm chosen as the SOTA benchmark in your experiments? Which specific linear search algorithm was used, and is it the one from Snell et al. (2024)? Additionally, why were no comparisons made with the works mentioned in the related work section? Please clarify the rationale behind these choices, as the novelty and contributions of this work are unclear without this context.

---

> ### Author Response · Authors · 2024-11-21
> **Response to Reviewer QZQj [1/1]**
>
> We thank the reviewer for their comments and for engaging with the paper. To address the concerns, we add new results showing the efficacy of backtracking in the code contests domain, clarify the linear search algorithm, and updated the paper for clarity. We are happy to clarify any questions you may have. Please let us know if your concerns are addressed and if so we would be grateful if you would be willing to raise your score. We would be happy to discuss if you have any concerns.
>
> > The dataset is limited to MATH domain.
> We have added CodeContests, as an additional dataset to show the efficacy of our approach on competitive programming, complementing our results in MATH (Hendryks et al, 2023), which studies high school math problems. In this task, an LLM must write code for a set of contest coding problems from a variety of sources (e.g CodeForces, HackerEarth, and more). To formulate code generation as an MDP, we define a step/action as a line of code, and the state/context as the concatenation of the contest problem (consisting of the problem description, runtime requirements, and sample input/output pairs) and the code so far. Evaluation is done with a python execution server on a set of held out contest problems. We present [preliminary results](https://imgur.com/code-contests-oracle-verifier-y5qR14t) with an Oracle Verifier, comparing our backtracking algorithm (orange) with Best-of-N (Blue, Snell et al. 2024) and Backtracking to a random step (Green).
>
> We find an average improvement of 10.16% from the linear search algorithm of Best-of-N, Snell et al. 2024 (backtracking from the first step) and significantly outperform backtracking to a random step. Note this is preliminary and we hope to train a PRM with Monte-Carlo Rollouts by the end of the rebuttal period or for the final version of the paper.
>
> > Why was a linear search algorithm chosen as the SOTA benchmark in your experiments? Which specific linear search algorithm was used, and is it the one from Snell et al. (2024)?
>
> In Figure 5, backtracking to the first step corresponds to Best-of-N Sampling with a PRM (Snell et al, 2024, Charniak & Johnson, 2005).  We want to clarify that our comparisons consist of the most prevalent search algorithms typically used in conjunction with PRMs to the best of our knowledge. In particular, the linear search algorithm consists of Best-of-N and it is the one from Snell et al. 2024. Note we make use of parallel sampling in our backtracking algorithm after we localize the incorrect step with the PRM (Section 3.2 - Suffix Generation and Stopping Criteria). Thus, for comparison, the choice of the parallel sampling algorithm is consistent for baselines and the backtracking approach. For works that alter the proposal distribution (RISE, SCoRe, and Self-Refine), these works are complementary to our work as our proposal distribution can be substituted with the modified learned distributions presented in these works. We hope to add additional linear search approaches such as beam search for the final iteration of the paper.
>
> > The writing needs to be improved.
>
> Thank you for the feedback, we have uploaded an updated version of the paper for your review. Please let us know if you have any additional suggestions or comments about this?

---

> > ### Author Response · Authors · 2024-11-24
> > **Following up!**
> >
> > Thank you for your review! Please let us know if further detail is needed or if the new experiments address your concerns.

---

> > > ### Author Response · Authors · 2024-11-27
> > > **Following up again!**
> > >
> > > Thank you again for your valuable feedback! With the discussion period ending soon, we kindly ask if the additional experiments and clarifications provided address your concerns or if there are any remaining points we can clarify before the deadline.

---

### Official Review · Reviewer_PD42 · 2024-11-05

**Soundness:** 2
**Presentation:** 3
**Contribution:** 3
**Rating:** 6
**Confidence:** 4

**Summary:**

The problem focuses on solving reasoning problems using LLMs which typically involves complex, multi-step generation where small mistakes in early steps can ultimately lead to incorrect outputs. While previous approaches have focused on linear search in parallel using a fixed budget, the proposed approach is non-linear and based on backtracking using learned process-based verifiers which allows localizing errors and proposing target revisions for these errors. Experiments show improvement of 15% over linear search algorithms.

**Strengths:**

Strength:
- Interesting and novel approach utilizing backtracking to improve test-time generation efficiency
- Paper is mostly clear and written well
- Experiments show promising results for the proposed approach

**Weaknesses:**

Weaknesses:
- I could not find clear comparison to linear search approaches like beam search, Best of N, etc., e.g., in terms of performance vs. budget. For example, we can compare a large number of candidates generated in parallel vs. additional turns; the paper explains why this may be inefficient in Section 3, but I did not see experiments validating this claim. Also, it was not clear to me how the 15% improvement is computed.

- I found parts of Algorithm 1 to be a bit confusing. It takes as an input "Advantage threshold" and "Revision margin t", however: (1) advantage threshold uses the same symbol as the initial trajectory; (2) both do not seem to be used in the algorithm; (3) I imagine one of them corresponds to the "Advantage smoothing for effective backtrack step selection" but it is not clear which one of them.

- Limited technical novelty: while the concept of backtracking to the correct step is novel, most elements including the PRM that provides per step estimates and its training have been adopted from previous literature (Snell et al., Wang et al.)

- Experimental setup could be extended to evaluate performance on other reasoning tasks and additional LMs (currently using Llama 8B, would be interesting to see performance on additional, larger models)

Minor:
Acronyms: "PRM" appears in abstract before definition, "SFT" in appendix without definition

**Questions:**

See point under "weaknesses" above. In particular, I would appreciate clarification regarding the comparison with linear search algorithm.

---

> ### Author Response · Authors · 2024-11-21
> **Response to Reviewer PD42 [1/2]**
>
> We thank the reviewer for their comments and for engaging with the paper. To address the concerns, we add new results showing that our approach is effective on coding problems from the CodeContests dataset. We are happy to clarify any questions you may have. Please let us know if your concerns are addressed and if so we would be grateful if you would be willing to raise your score. We would be happy to discuss if you have any concerns.
>
> > I found parts of Algorithm 1 to be a bit confusing. It takes as an input "Advantage threshold" and "Revision margin t", however: (1) advantage threshold uses the same symbol as the initial trajectory; (2) both do not seem to be used in the algorithm; (3) I imagine one of them corresponds to the "Advantage smoothing for effective backtrack step selection" but it is not clear which one of them.
>
> We have updated Algorithm 1 in the new revision of the paper (for your [reference](https://imgur.com/gallery/backtracking-algorithm-description-QMAvAQC)).The advantage threshold, referred now by $\omega$, is used and we keep the initial solution trajectory as $\tau$. We have removed the revision margin $t$ for clarity and added the stopping criterion.
>
> To walk through the psuedocode, we start with an input trajectory $\tau = \{s_1, s_2, \dots, s_N\}$. Next, we compute an advantage function $A(s_i, a_i)$ for each step within the trajectory. In each iteration, we identify the step $s_{i_{\text{revise}}}$ with the lowest advantage. To allow for estimation errors, we smooth the advantage function and choose a more conservative step, $i_{\text{smooth}}$, to backtrack to, increasing the likelihood that the step we bactrack to and its previous steps are correct. Then, the trajectory is then resampled starting from the step $T=i_{\text{smooth}} - 1$ by using a linear search algorithm (e.g., Best-of-N) as a subroutine. If the new trajectory $\tau{\prime}$ improves the solution, we update the best trajectory seen so far $\tau_{\text{best}}$. This process is repeated until either a maximum number of iterations M is reached or the stopping criterion is met (using the PRM’s value at the final step of the trajectory to evaluate correctness). The algorithm finally outputs the revised trajectory $\tau_{\text{best}}$, representing the most improved version of the input trajectory.
>
> Please let us know if there is anything more we can do to improve the clarity of the algorithm.
>
> > Limited technical novelty: while the concept of backtracking to the correct step is novel, most elements including the PRM that provides per step estimates and its training have been adopted from previous literature (Snell et al., Wang et al.)
>
> We would like to clarify that the main contribution of our work is in showing that sequential improvements on a response when backtracking with a PRM and running sequential sampling. Concretely, we summarize our contribution as follows:
>
> 1. **Sequential improvement by backtracking**: We propose a novel approach to sequentially improve model reasoning by using the advantage function to identify the incorrect step and revising our solution from there. This helps, both, avoid excessive compute used by best-of-N and beam search in re-generating the correct part of the response and continuing after a fatal error. This improves, both computational efficiency and accuracy of the search.
>
> 2. **Improving the value function calibration**: We propose multiple approaches for improving the value function calibration for backtracking. These include 1.) Using an in-context verifier to reduce uncertainty of the value function on future revisions 2.) fine-tuning with a balanced dataset to deal with the dataset imbalances arising from backtracking and 3.) smoothing of the advantage function and conservatively choosing a step to revise to, ensuring that the step is in a correct portion of the solution.
>
> Our approach provides substantial improvements in terms of performance as a function of inference-compute budget (Figure 5 + Figure 1 Rebuttal, CodeContests), which we believe should be of value and interest to the community interested in LLMs and scaling test-time compute.  We additionally show that given a small inference token budget, we can obtain the best response.
>
> We do note that Section 3.5 does include a set of practical considerations of learning a PRM a portion of which has been adopted from previous literature primarily because other details are needed to train an effective verifier. However, other practical conditions we address (such as issues with dataset balance for backtracking as well as smoothing of the advantage function) are additional novel contributions. We do not claim that learning the verifier itself is novel but rather using it in this way for search is. For completeness, we still wanted to identify and communicate details that helped the training of value verifiers in the context of the backtracking algorithm.

---

> ### Author Response · Authors · 2024-11-21
> **Response to Reviewer PD42 [2/2]**
>
> > Experimental setup could be extended to evaluate performance on other reasoning tasks
>
> We have added CodeContests, as an additional dataset to show the efficacy of our approach on competitive programming, complementing our results in MATH (Hendryks et al, 2023), which studies high school math problems. In this task, an LLM must write code for a set of contest coding problems from a variety of sources (e.g CodeForces, HackerEarth, and more). To formulate code generation as an MDP, we define a step/action as a line of code, and the state/context as the concatenation of the contest problem (consisting of the problem description, runtime requirements, and sample input/output pairs) and the code so far. Evaluation is done with a python execution server on a set of held out contest problems. We present [preliminary results](https://imgur.com/code-contests-oracle-verifier-y5qR14t) with an Oracle Verifier, comparing our backtracking algorithm (orange) with Best-of-N (Blue, Snell et al. 2024) and Backtracking to a random step (Green).
>
> We find an average improvement of 10.16% from the linear search algorithm of Best-of-N, Snell et al. 2024 (backtracking from the first step) and significantly outperform backtracking to a random step. Note this is preliminary and we hope to train a PRM with Monte-Carlo Rollouts by the end of the rebuttal period or for the final version of the paper.
>
> > I could not find clear comparison to linear search approaches like beam search, Best of N, etc., e.g., in terms of performance vs. budget.
>
> In Figure 5, backtracking to the first step corresponds to Best-of-N Sampling with a PRM (Snell et al, 2024, Charniak & Johnson, 2005).  We want to clarify that our comparisons consist of the most prevalent search algorithms typically used in conjunction with PRMs to the best of our knowledge. In particular, the linear search algorithm consists of Best-of-N and it is the one from Snell et al. 2024. Note we make use of parallel sampling in our backtracking algorithm after we localize the incorrect step with the PRM (Section 3.2 - Suffix Generation and Stopping Criteria). Thus, for comparison, the choice of the parallel sampling algorithm is consistent for baselines and the backtracking approach. For works that alter the proposal distribution (RISE (Qu et al 2024), SCoRe (Kumar et al 2024), and Self-Refine (Madaan et al, 2023)), these works are complementary to our work as our proposal distribution can be substituted with the modified learned distributions presented in these works as stated in the related works section. We hope to add additional linear search approaches such as beam search for the final iteration of the paper.
>
> > Also, it was not clear to me how the 15% improvement is computed.
>
> We use the accuracy vs tokens plot in Figure 5 for the learnt PRM to compute the average percent improvement over tokens of Backtracking with the Multi-Turn Value Function over the best performing baseline, Baseline 1: Backtracking from the First Step (Best-of-N). Therefore, the Multi-Turn Value Function has approximately a 15% improvement from Baseline 1. We will modify the text to avoid this confusion.
>
> > [...] additional LMs (currently using Llama 8B, would be interesting to see performance on additional, larger models)
>
> Due to compute constraints, it would be difficult to evaluate with additional larger models in time for the rebuttal. We plan to run evaluation with more models for the final iteration of the paper. Note in prior work (e.g Snell et al 2024, Setlur et al 2024) a single PaLM/Gemma model was utilized for the entirety of the experiments they present in their work. One additional evaluation we hope to run is an oracle verifier with different models for the code contests domain by the end of the rebuttal period or for the final version of the paper.

---

> > ### Author Response · Authors · 2024-11-24
> > **Following up!**
> >
> > Thank you for your review! Please let us know if further detail is needed or if the new experiments address your concerns.

---

> > > ### Author Response · Authors · 2024-11-27
> > > **Following up again!**
> > >
> > > Thank you again for your valuable feedback! With the discussion period ending soon, we kindly ask if the additional experiments and clarifications provided address your concerns or if there are any remaining points we can clarify before the deadline.

---

> > > > ### Comment · Reviewer_PD42 · 2024-11-27
> > > >
> > > > Thank you for your response and for providing additional experiments. I have no additional questions.

---

> > > > > ### Author Response · Authors · 2024-11-27
> > > > >
> > > > > We would like to express our sincere thanks and appreciation for your feedback. We would be most grateful if you would consider upgrading your score, given that your concerns have now been addressed.

---

### Official Review · Reviewer_YkYW · 2024-11-05

**Soundness:** 3
**Presentation:** 2
**Contribution:** 2
**Rating:** 5
**Confidence:** 2

**Summary:**

This paper aims to improve the efficiency of reasoning during inference. Specifically, this paper first uses process verifiers to predict likelihoods of success per step (for example using the process reward model to estimate the value function), which can be used to identify a problematic step. This significantly reduces the amount of computation. To further enhance the computational efficiency of inference, the authors introduce in-context process supervision.

**Strengths:**

(1) This proposed method improve the computational efficiency if one can use process reward model to get a good value function estimator and identify the problematic step correctly. And the empirical result in mathematical reasoning validate this.

(2) This method works well for multi-step tasks.

**Weaknesses:**

(1) The performance of the backtracking search method is largely dependent on the reward shaping methods, i.e., process reward model (PRM).  If the estimated rewards (and value functions) is not accurate enough, it will result in great challenges in backtracking search process. As the true reward is sparse (only 0,1 in the final), it seems not easy to estimate the process reward well.

(2) The authors evaluate how backtracking would perform on reasoning problems (Hendrycks Math dataset). Would this generalize well to other types of data or reasoning tasks?

Minors:

In the abstract and introduction, the abbreviation "PRM" appears before its full term is introduced, it is a little confusing.

**Questions:**

See above

---

> ### Author Response · Authors · 2024-11-21
> **Response to Reviewer YkYW [1/1]**
>
> We thank the reviewer for their comments and for engaging with the paper. To address the concerns, we add new results showing that our approach is effective on coding problems from the CodeContests dataset. We are happy to clarify any questions you may have. Please let us know if your concerns are addressed and if so we would be grateful if you would be willing to raise your score. We would be happy to discuss if you have any concerns.
>
>
> > (1) The performance of the backtracking search method is largely dependent on the reward shaping methods, i.e., process reward model (PRM). If the estimated rewards (and value functions) is not accurate enough, it will result in great challenges in backtracking search process. As the true reward is sparse (only 0,1 in the final), it seems not easy to estimate the process reward well.
>
> Note, that the underlying assumption for using verification is that there exists an insignificant gap between the generator and the verifier. This clearly exists in domains such as Math, Programming, Puzzles, and more. Thus, even if noisy, learning a PRM allows for some benefit when inference-time computation is utilized in tandem with it.
>
> The accuracy of the verifier is indeed important for the algorithm's success. However, all search-based approaches that leverage a verifier (e.g Snell et. al) have this inherent requirement. Below, we consider the simpler search algorithm of Best-of-N. Here, we artificially add label noise to the Oracle verifier and see what the performance of the algorithm looks like for a subset of the math dataset. As seen in the following [plot](https://imgur.com/gallery/QXt1hN8), as the verifier accuracy decreases, there is a large drop in the performance of a simpler search algorithm such as Best of N. Thus, given that this is a limitation for prior approaches, we should not be penalized for needing a verifier, with sufficient accuracy.
>
> For this reason, we discuss in detail what it takes to practically learn such a verifier. For example, we outline in Section 3.5 how to practically learn a PRM to attain a higher verification accuracy. Additionally, we propose in-context value verification in Section 3.4 to further reduce the uncertainty of the verifier when performing revisions. Finally, we smooth the verifier during backtracking as seen in section 4.2 to be conservative with the step we backtrack to, ensuring that the step chosen to restart generation is indeed in a correct portion of the previous solution. This reduces the inefficiencies that come from backtracking to an incorrect step, and the smoothness can be relaxed given a stronger verifier.
>
>
> > (2) The authors evaluate how backtracking would perform on reasoning problems (Hendrycks Math dataset). Would this generalize well to other types of data or reasoning tasks?
>
> We have added CodeContests, as an additional dataset to show the efficacy of our approach on competitive programming, complementing our results in MATH (Hendryks et al, 2023), which studies high school math problems. In this task, an LLM must write code for a set of contest coding problems from a variety of sources (e.g CodeForces, HackerEarth, and more). To formulate code generation as an MDP, we define a step/action as a line of code, and the state/context as the concatenation of the contest problem (consisting of the problem description, runtime requirements, and sample input/output pairs) and the code so far. Evaluation is done with a python execution server on a set of held out contest problems. We present [preliminary results](https://imgur.com/code-contests-oracle-verifier-y5qR14t) with an Oracle Verifier, comparing our backtracking algorithm (orange) with Best-of-N (Blue, Snell et al. 2024) and Backtracking to a random step (Green).
>
> We find an average improvement of 10.16% from the linear search algorithm of Best-of-N, Snell et al. 2024 (backtracking from the first step) and significantly outperform backtracking to a random step. Note this is preliminary and we hope to train a PRM with Monte-Carlo Rollouts by the end of the rebuttal period or for the final version of the paper.

---

> > ### Author Response · Authors · 2024-11-24
> > **Following up!**
> >
> > Thank you for your review! Please let us know if further detail is needed or if the new experiments address your concerns.

---

> > > ### Author Response · Authors · 2024-11-27
> > > **Following up again!**
> > >
> > > Thank you again for your valuable feedback! With the discussion period ending soon, we kindly ask if the additional experiments and clarifications provided address your concerns or if there are any remaining points we can clarify before the deadline.

---

### Official Review · Reviewer_RBMD · 2024-11-05

**Soundness:** 3
**Presentation:** 3
**Contribution:** 3
**Rating:** 5
**Confidence:** 2

**Summary:**

This work focuses on the inference acceleration of LLMs by using a backtracking method when the LLMs outputs the inference intermediate steps. The authors propose to detect errors in intermediate reasoning steps using PRMs and resamples the incorrect part of a solution, saving computational resources. The authors also introduce the so-called in-context process supervision, where the model adjusts verification based on past attempts. This further reduces the computation of LLMs.

**Strengths:**

1. This paper proposes new methods to localize the incorrect part of the responses generated by the LLMs. The proposed in-context process supervision adapts verification based on historical data, which improves the performance of LLMs inference.

2. The efficacy of the proposed methods are verified on the mathematical problems.

**Weaknesses:**

1. The experiments of the work mainly focus on mathematical problems. However, the applicability of the proposed algorithm to other domains, e.g., code and legal reasoning, remains untested.

2. The efficacy of the proposed algorithm heavily depends on the accuracy of PRM. The incorrect error localization could lead to inefficient backtracking and degenerate performance. The influence of the PRM accuracy is not well-explored in the current work.

**Questions:**

Same as the Weaknesses part.

---

> ### Author Response · Authors · 2024-11-21
> **Response to Reviewer RBMD [1/1]**
>
> We thank the reviewer for their comments and for engaging with the paper. To address the concerns, we add new results showing that our approach is effective on coding problems from the CodeContests dataset. We are happy to clarify any questions you may have. Please let us know if your concerns are addressed and if so we would be grateful if you would be willing to raise your score. We would be happy to discuss if you have any concerns.
>
> > The experiments of the work mainly focus on mathematical problems. However, the applicability of the proposed algorithm to other domains, e.g., code and legal reasoning, remains untested.
>
> We have added CodeContests, as an additional dataset to show the efficacy of our approach on competitive programming, complementing our results in MATH (Hendryks et al, 2023), which studies high school math problems. In this task, an LLM must write code for a set of contest coding problems from a variety of sources (e.g CodeForces, HackerEarth, and more). To formulate code generation as an MDP, we define a step/action as a line of code, and the state/context as the concatenation of the contest problem (consisting of the problem description, runtime requirements, and sample input/output pairs) and the code so far. Evaluation is done with a python execution server on a set of held out contest problems. We present [preliminary results](https://imgur.com/code-contests-oracle-verifier-y5qR14t) with an Oracle Verifier, comparing our backtracking algorithm (orange) with Best-of-N (Blue, Snell et al. 2024) and Backtracking to a random step (Green).
>
> We find an average improvement of 10.16% from the linear search algorithm of Best-of-N, Snell et al. 2024 (backtracking from the first step) and significantly outperform backtracking to a random step. Note this is preliminary and we hope to train a PRM with Monte-Carlo Rollouts by the end of the rebuttal period or for the final version of the paper.
>
> > The efficacy of the proposed algorithm heavily depends on the accuracy of PRM. The incorrect error localization could lead to inefficient backtracking and degenerate performance. The influence of the PRM accuracy is not well-explored in the current work.
>
> Note, that the underlying assumption for using verification is that there exists an insignificant gap between the generator and the verifier. This clearly exists in domains such as Math, Programming, Puzzles, and more. Thus, even if noisy, learning a PRM allows for some benefit when inference-time computation is utilized in tandem with it.
>
> The accuracy of the verifier is indeed important for the algorithm's success. However, all search-based approaches that leverage a verifier (e.g Snell et. al) have this inherent requirement. Below, we consider the simpler search algorithm of Best-of-N. Here, we artificially add label noise to the Oracle verifier and see what the performance of the algorithm looks like for a subset of the math dataset. As seen in the following [plot](https://imgur.com/gallery/QXt1hN8), as the verifier accuracy decreases, there is a large drop in the performance of a simpler search algorithm such as Best of N. Thus, given that this is a limitation for prior approaches, we should not be penalized for needing a verifier, with sufficient accuracy.
>
> For this reason, we discuss in detail what it takes to practically learn such a verifier. For example, we outline in Section 3.5 how to practically learn a PRM to attain a higher verification accuracy. Additionally, we propose in-context value verification in Section 3.4 to further reduce the uncertainty of the verifier when performing revisions. Finally, we smooth the verifier during backtracking as seen in section 4.2 to be conservative with the step we backtrack to, ensuring that the step chosen to restart generation is indeed in a correct portion of the previous solution. This reduces the inefficiencies that come from backtracking to an incorrect step, and the smoothness can be relaxed given a stronger verifier.

---

> > ### Author Response · Authors · 2024-11-24
> > **Following up!**
> >
> > Thank you for your review! Please let us know if further detail is needed or if the new experiments address your concerns.

---

> > > ### Author Response · Authors · 2024-11-27
> > > **Following up again!**
> > >
> > > Thank you again for your valuable feedback! With the discussion period ending soon, we kindly ask if the additional experiments and clarifications provided address your concerns or if there are any remaining points we can clarify before the deadline.

---

### Official Review · Reviewer_vxtv · 2024-11-05

**Soundness:** 2
**Presentation:** 1
**Contribution:** 2
**Rating:** 3
**Confidence:** 2

**Summary:**

The paper presents a new method for improving the reasoning capabilities of large language models. The reasoning process can be viewed as a search process through a sequence of steps with each step logically building upon the previous ones until the desired conclusion is reached. This process can be formally modelled as a Markov Decision Process and solved via reinforcement learning. A key component of the solving approach is a process verifier that can predict the likelihood of success per step and thus facilitate backtracking to maximise performance per generated token. These process verifiers can be learned from relatively small datasets. The empirical evaluation seems to indicate that the proposed approach improves the reasoning process by employing preemptive backtracks.

**Strengths:**

- Improving the reasoning capabilities of existing language models is definitely a topic of considerable interest in the AI community. The proposed approach seems to address this issue in a principled manner.

**Weaknesses:**

- The presentation is quite dense and therefore it is not easy to follow the details of the paper. The example from Figure 2 is nice and helps to understand the basic idea. However, Sections 3.2 to 3.5 which contain the main contribution are not easy to understand. Perhaps including more illustrative examples here would improve the quality of the presentation.

- The empirical evaluation seems to be limited to a single dataset. In addition, it is hard to understand what the plots in Figure 7 present.

**Questions:**

- Is it possible to do some kind of transfer learning, namely learning a process verifier in one domain and either use it directly or fine-tuning for another domain?

- What language models were used in the evaluation?

---

> ### Author Response · Authors · 2024-11-21
> **Response to Reviewer vxtv [1/1]**
>
> We thank the reviewer for their comments and feedback, especially for agreeing that our method approaches improving the reasoning capabilities of language models in a principled manner. To address the main concerns, we have now updated the paper to improve clarity of Sections 3.2 and 3.5 and explain them briefly below. In addition, we add new experimental results showing that our approach for backtracking is effective on coding problems. Please let us know if your concerns are addressed and if so we would be grateful if you would be willing to raise your score. We would be happy to discuss if you have any concerns.
>
> > The empirical evaluation seems to be limited to a single dataset.
>
> We have added CodeContests, as an additional dataset to show the efficacy of our approach on competitive programming, complementing our results in MATH (Hendryks et al, 2023), which studies high school math problems. In this task, an LLM must write code for a set of contest coding problems from a variety of sources (e.g CodeForces, HackerEarth, and more). To formulate code generation as an MDP, we define a step/action as a line of code, and the state/context as the concatenation of the contest problem (consisting of the problem description, runtime requirements, and sample input/output pairs) and the code so far. Evaluation is done with a python execution server on a set of held out contest problems. We present [preliminary results](https://imgur.com/code-contests-oracle-verifier-y5qR14t) with an Oracle Verifier, comparing our backtracking algorithm (orange) with Best-of-N (Blue, Snell et al. 2024) and Backtracking to a random step (Green).
>
> We find an average improvement of 10.16% from the linear search algorithm of Best-of-N, Snell et al. 2024 (backtracking from the first step) and significantly outperform backtracking to a random step. Note this is preliminary and we hope to train a PRM with Monte-Carlo Rollouts by the end of the rebuttal period or for the final version of the paper.
>
> > It is hard to understand what the plots in Figure 7 present.
>
> In this figure, we show how backtracking can allow for optimization of the value function through subsequent revisions. The goal of backtracking is to reset to the portion of the solution right before the model went on the wrong track. With our approach, we are able to identify this reset point with the advantage,( computed as the delta in values between subsequent reasoning steps). We identify the backtracking step as one with the maximum negative delta with its subsequent step, i.e., one with minimum advantage. For example, in the 1st revision, we found that the lowest advantage is in Step 5, which corresponds to the biggest drop in the value function from step 5 to step 6. When we resample trajectories from this poor step (Step 5), we find that the new trajectory generated has higher values. Looking over multiple revisions, we see that the value function is monotonically increasing over the course of the revisions, which is a desirable property of the backtracking algorithm. Note that the value function corresponds to the probability of success of a particular state (context) and action (step) pair, i.e, a higher value at a step indicate higher chances of generating successful trajectory. Hence, increasing the value function should lead to a higher accuracy for a set of problems. We will update the paper to explain this better..
>
> > The presentation is quite dense and therefore it is not easy to follow the details of the paper.
>
> We have updated the paper and have added clarity as you have suggested in the preliminaries and method section of the paper. We would be happy to clarify any questions you may have that leads to a better understanding of this paper.
>
> > What language models were used in the evaluation?
>
> For our experiments, we use the Meta Llama-3.1-8B-Instruct models.

---

> > ### Author Response · Authors · 2024-11-24
> > **Following up!**
> >
> > Thank you for your review! Please let us know if further detail is needed or if the new experiments address your concerns.

---

> > > ### Author Response · Authors · 2024-11-27
> > > **Following up again!**
> > >
> > > Thank you again for your valuable feedback! With the discussion period ending soon, we kindly ask if the additional experiments and clarifications provided address your concerns or if there are any remaining points we can clarify before the deadline.

---

> > ### Comment · Reviewer_vxtv · 2024-11-27
> >
> > Thanks for the clarifications.

---

> > > ### Author Response · Authors · 2024-11-27
> > >
> > > We sincerely appreciate your valuable feedback and the time you have dedicated to reviewing our work. Could you kindly let us know if there are any remaining reservations regarding the acceptance of the paper? Given the extension in the rebuttal period, we would be happy to address them. If all concerns have been addressed to your satisfaction, we would be truly grateful if you could upgrade your score.

---

### Author Response · Authors · 2024-12-03
**Overall response to All Reviewers**

We sincerely thank the reviewers for their constructive feedback and insightful suggestions, which have significantly improved our paper. Below, we summarize the major changes and address specific comments in individual responses.

**Key Updates:**

- *Expanded Empirical Evaluation on CodeContests*
(vxtv, rbmd, ykyw, pd42, QZQJ)
We added results using the oracle verifier on CodeContests, demonstrating the efficacy of backtracking. Additionally, we included [results](https://imgur.com/a/CLTTIAD) with a stronger model (Yi-Coder-9B-chat), showing that the findings extend beyond the Llama family of models. We will include the full set of results with a learnt PRM for the final revision of the paper.
- *Addressing Perceived Limitations Due to PRM Accuracy*
(vxtv, rbmd, ykyw)
We conducted an [additional experiment](https://imgur.com/gallery/QXt1hN8) highlighting how even simpler search algorithms, such as Best-of-N (leveraged in prior work such as Snell et al 2023), are affected by inaccuracies in the PRM. We also discussed practical considerations for learning an effective PRM in section 4.2 for error localization and search and proposed in-context value verification in section 4.3 to reduce verifier uncertainty during subsequent revisions.
- *Latency*
(1tMN)
We measure the generation latency by comparing sequential revisions to parallel sampling. Despite seeing an increase in latency due to the sequential generation, our approach leads to a favorable tradeoff between performance (accuracy) and FLOPs, an established metric for computation cost as used in prior work such as Sardana et al. (2023). Additionally, both prior academic work such as RISE (Yu et al, 2024) and Snell et al 2024 as well as SOTA reasoning systems such as O1-preview, Deepseek r1-lite, and Qwen QWQ, similarly incur latency trade-offs due to sequential sampling aimed at improving the performance of the overall reasoning system. Therefore, this is a metric we shouldn’t be penalized for.
- *Improved Clarity Through Rewriting*
(general)
We significantly rewrote Sections 3 and 4 to enhance clarity and better articulate the contributions of our method. We are open to making further revisions to refine the paper’s writing and presentation.

---

### Meta-Review · Area_Chair_LKVZ · 2024-12-17

**Metareview:**

The paper proposes to improve efficiency of test-time search in LLMs with backtracking based on PRMs. This reasoning process of LLMs can be formally modelled as a Markov Decision Process and solved via reinforcement learning. A key component of the solving approach is a process verifier that can predict the likelihood of success per step and thus facilitate backtracking to maximise performance per generated token.

In general, reviewers felt that the work has some merit, as it localizes the incorrect parts of LLMs' outputs. Experiments on Math datasets are also illuminating.

However, reviewers were unanimous in their assessment that this paper doesn't cross the bar for ICLR. Let me summarize some of their reservations.

1) The experiments of the work mainly focus on mathematical problems. This is a major limitation in view of the broad applicability of LLMs.
2) The exposition of the paper and presentation of the ideas in the paper were not ideal, making it hard to comprehend. I acknowledge that this improved during the course of the discussion, but feel that the discussion phase is not the right point to refine the paper. It requires another round of thorough review.
3) Limited technical novelty: while the concept of backtracking to the correct step is novel, most elements including the PRM that provides per step estimates and its training have been adopted from previous literature (Snell et al., Wang et al.).
4) Comparison to linear search is inadequate. This is was mentioned by multiple reviewers.
5) Language models used are not sufficiently large.

Personally, I am not an expert at the topic. Hence, my assessment is based on the reviewers' feedback, which is unanimous. I acknowledge the authors' message to me regarding their objections to Reviewer 1TMN viewpoint regarding the applicability of linear search in this setting. However, this point is mentioned several times by various reviewers. Thus, I cannot ignore it and encourage the authors to take this comment seriously into consideration in their comparisons to the baselines in future work. At present, the work is premature and does not pass the bar for ICLR.

**Additional Comments On Reviewer Discussion:**

The authors provided detailed responses to the reviewers' comments, conducting additional experiments among other responses. However, reviewers remained unconvinced. Reviewer 1TMN even increased their confidence, underscoring the fact that they feel more confident about their point after the authors' responses.

I took their comments, scores, and confidence scores into account in coming to my decision.

---

### Decision · Program_Chairs · 2025-01-22

Reject